# A Snapshot of Knowledge about Oral Cancer in Italy: A 505 Person Survey

**DOI:** 10.3390/ijerph17134889

**Published:** 2020-07-07

**Authors:** Riccardo Nocini, Giorgia Capocasale, Daniele Marchioni, Francesca Zotti

**Affiliations:** 1Department of Otorhinolaryngology—Head and Neck Surgery, University Hospital of Verona, 37126 Verona, Italy; riccardo.nocini@gmail.com (R.N.); daniele.marchioni@univr.it (D.M.); 2Section of Dentistry and Maxillofacial Surgery, Department of Surgical Sciences, Paediatrics and Gynaecology, University of Verona, 37134 Verona, Italy; francesca.zotti@univr.it

**Keywords:** oral cancer, diagnostic delay, patients awareness

## Abstract

*Objectives*: Patients’ knowledge about oral squamous cell carcinoma (OSCC) plays an important role in primary prevention, early diagnosis, and prognosis and survival rate. The aim of this study was to assess OSCC awareness attitudes among general population in order to provide information for educational interventions. *Methods*: A survey delivered as a web-based questionnaire was submitted to 505 subjects (aged from 18 to 76 years) in Italy, and the answers collected were statistically analyzed. Information was collected about existence, incidence, features of lesions, risk factors of oral cancer, and self-inspection habits, together with details about professional reference figures and preventive behaviors. *Results*: Chi-square tests of independence with adjusted standardized residuals highlighted correlations between population features (age, gender, educational attainment, provenance, medical relationship, or previous diagnoses of oral cancer in family) and knowledge about oral cancer. *Conclusions*: Knowledge about OSCC among the Italian population is limited, and it might be advisable to implement nudging and sensitive customized campaigns in order to promote awareness and therefore improve the prognosis of this disease.

## 1. Introduction

Oral squamous cell carcinoma (OSCC) is the most common malignancy of the oral cavity; indeed, 90% of cancers in the dental area originate histologically in the squamous cells [1]. Data from GLOBOCAN 2018 were used to estimate that about 4500 new cases and 1500 deaths from OSCC occurred in 2018 in Italy [2]. 

The risk factors are well-known: smoking [3] and alcohol [4], which are considered modifiable major risk factors; others include the human papillomavirus (HPV) [5,6], poor oral hygiene [7], and chronic injuries of the mucosa [8]. Moreover, OSCC is easy detectable [9], thanks to the ease of inspection of the oral cavity and the frequent previous occurrence of oral potentially malignant disorders (OPMDs) including leukoplakia and erythroplakia, which are the most common ones [1].

Early detection plays a key role in achieving the best prognosis and long survival in cases of OSCC; this issue is a relevant global health problem, especially since the 5 year survival for patients has not improved significantly in the last few decades, remaining below 50% [2]. 

This trend could be explained in terms of diagnostic delay of OSCC. It is reported in the literature that two thirds of OSCC cases are diagnosed at advanced stages (Stage III or IV) [10]; therefore, prognosis and survival rate become serious concerns. 

Diagnostic delay of OSCC can be distinguished into “patient delay”, which is the period between the patients’ first detection and seeking healthcare, and “professional delay”, which is the period from the first examination to the final diagnosis of the condition [10]. Hence, the main cause of death in these cases is patients’ and physicians’ lack of awareness and knowledge about this condition.

It is reported in the literature that 58% of delays were caused by patients’ procrastinating in consulting health professionals [11], and that this could result in about 1.6–5.4 months of diagnostic delay [10]. This deferral is strongly related to patients’ lack of knowledge about cancer issues. Indeed, in Stages I and II, OSCC lesions may cause no discomfort; therefore, the patient does not consult the dentist or other specialists and the pathological condition becomes more serious and difficult to treat [12]. Patients perceive signs and symptoms as being related to minor oral disease, e.g., trauma, infective process, disorders related to dentures, or other generic, non-dangerous dental conditions. As a consequence, self- and/or inappropriate medications are frequently applied. Thus an effective strategy to reduce OSCC diagnostic delay should address these factors, which primarily involve cognitive processes in the initial stages of disease [13]. 

All such strategies have to be taken up by the population and, for this reason, an assessment of knowledge and a consequent educational intervention could be the action that makes the difference between survival and mortality, or between good and poor quality of life.

The aim of this study was to assess oral cancer awareness attitudes among the Italian general population in order to provide up-to-date information and, hopefully, to suggest educational interventions based on the results.

## 2. Materials and Methods

This cross-sectional survey was conducted between June 2019 and January 2020 among a random sample of volunteers in Italy. No incentive was offered to participants.

An anonymous web-based questionnaire in Italian was shared to the Italian-speaking sample by mean of e-mail, WhatsApp™ (Facebook Inc., Mountain View, California, USA) and Telegram Messenger chat rooms (Telegram LLC, Dubai, United Arab Emirates), and Facebook (Facebook Inc., Menlo Park, CA, USA). It was created using Google Forms (Alphabet Co., Mountain View, CA, USA). The surveys were electronically answered and the answers were linked to the author’s Google account. To ensure data security, results were not made public. The survey required approximately 5 min to complete. It was completely anonymous and no sensitive questions were asked.

The questionnaire was sent to dental practitioners, asking them for sharing it with patients and to the general population by URL sharing, thus building on knowledge of all authors and their colleagues. 

The questionnaire consisted of two sections: the first one included closed-ended demographic questions (age, gender, educational attainment, relationship to people in the medical sector, region of provenance in Italy, belonging to medical/dental field); the second one presented closed and closed-ended questions investigating individual oral cancer awareness, related to epidemiology in Italy, risk factors, mouth self-examination, and signs and symptoms of OSCC. The following questions were included in the second part of questionnaire, and more than one answer was indicated as correct.
(1)Do you have family members/acquaintances diagnosed with oral cancer? Yes or no(2)Do you have relatives in dental or medical fields? Yes or no(3)Are you belonging to medical/dental field? Yes or no(4)Do you know the existence of cancer of the mouth and lips? Yes or no(5)As far as you are aware, how many cases of oral cancer are registered in Italy every year?(a)Fewer than 500(b)About 500(c)About 1500(d)About 3000(e)About 4500(f)About 6000(g)More than 10,000(6)Which of these factors are related to an increased risk of oral cancer?(a)Smoking habit(b)Alcohol consumption(c)Papilloma virus(d)Poor hygiene(e)All previous answers(7)Which of these features can be initial presentations of oral carcinoma?(a)White plaque(b)Reddish plaque(c)Spreading burning sensation in the mouth(d)Dry mouth(8)Do you know how to self-inspect your mouth? Yes or no(9)If you notice an ulcer on your tongue, how long would you wait before seeking medical attention?(a)2 weeks(b)2 months(c)4 months(10)Whom would you seek out to check it?(a)General practitioner(b)Dentist(c)Otolaryngologist(d)Anyone(e)I do not know(11)How often do you visit the dentist?(a)Twice a year(b)Once a year(c)Once in two years(d)Less than once every three years

The study protocol and the questionnaire were approved by the review board of Dental School of Verona (University of Verona, Italy). Ethical committee approval was not sought because the respondents and their personal data were protected by anonymity. Not even the person who analyzed the data could link information to a particular person; furthermore, people could not be recognized by answers given. The validity of a questionnaire is determined by analyzing whether the questionnaire measures what it is intended to measure. The content validity was assessed by a panel of researchers from the Dental School of Verona (review board) [14].

All questions were clear and unambiguous: a pilot test of the questionnaire was performed with a random sample of 10 patients referred to our department (on a volunteer basis) to ensure practicability, validity, and the interpretation of answers. The questions were then revised considering the comments obtained before the link of the web-based questionnaire was shared to the study sample.

Furthermore, the authors responsible for designing the study (NR, CG) checked critical points and managed them in order to make the questions clearer. Final approval was given by all authors before the questionnaire was submitted to the sample.

All data were recorded in a database and processed by only one operator (ZF).

No exclusion criteria were stated in submitting and sending the questionnaire. In collecting answers and analyzing data, only answers from subjects belonging to medical field were excluded in order to ensure a picture of laypeople and thus avoid selection bias.

Age groups were identified as follows: Group 1, aged between 18 and 35; Group 2, aged between 36 and 50; and Group 3, aged over 50.

### Ethical Considerations

All procedures performed in studies involving human participants were in accordance with the ethical standards of the institutional research committee (University of Verona, Italy) and with the 1964 Helsinki declaration and its later amendments or comparable ethical standards.

## 3. Statistical Analysis

Descriptive analysis was performed to evaluate
Demographic data of populationKnowledge about incidence of oral cancer in Italy

Chi-squared independence tests with adjusted standardized residual calculations were carried out to assess correlations between age, gender, educational attainment, region of provenance, connections with people in medical or dental fields, and acquaintances diagnosed with oral cancer and the following variables:existence of oral and lip cancers,risk factors,features of oral cancer lesions,habits of self-inspection,waiting time before seeking medical attention,professional reference figures,habits of routine dental visits.

In the cases of few observations, Fisher’s exact test was performed. 

All test were considered statistically significant for *p* ≤ 0.05.

All statistical analyses were performed using Statistical Package for Social Sciences Version 25.0 (SPSS Inc., Chicago, IL, USA).

## 4. Results

A total of 505 participants completed the web-based questionnaire. Table 1 presents the statistical analysis.

### 4.1. Demographic Data of Population

Table 2 presents the demographic characteristics of respondents.

The sample comprised a gender distribution of 71.49% females and 28.51% males. The ages of the participants ranged from 18 to 76 years, with 316 participants in the Group 1, 81 in Group 2, and 108 in Group 3. Of the participants recruited, 251 were from Northern Italy and 254 from the south. 

A total 44.75% of participants reported having a high-school diploma, 36.23% a graduate degree, 20% a junior-school diploma, and 1.38% a PhD; moreover, 153 participants reported having relatives in medical or dental fields. Additionally, 4.75% reported having acquaintances diagnosed with OSCC. 

### 4.2. Existence of Oral and Lip Cancers

Figure 1 presents the results: about 59.60% of the participants reported awareness of oral cancer, with important significant associations for gender (*p* = 0.001), age (*p* = 0.002 for Group 1), educational attainment (*p* = 0.0000 for junior high school and graduate degrees), subjects with connections in dental or medical fields, and acquaintances diagnosed with OSCC (*p* = 0.000); however, no significant association was found with region of provenance (*p* > 0.05). In particular, answers by interviewees with junior-school and graduate education showed a significant correlation (*p* = 0.0000).

### 4.3. Knowledge about Incidence of Oral Cancer in Italy

A total 68.12% of participants reported that every year, in Italy, <3000 new cases of OSCC are registered, and of these, 36.62% answered fewer than or about 500 (Figure 2).

### 4.4. Risk Factors

Figure 3 presents the results for questions on risk factors. Among all risk factors for OSCC, smoking was identified as the only risk factor by 18.61% of the participants; it was identified in all answers where more than one risk factor was reported (89.30% of the participants). However, only 58.01% of the participants identified alcohol as a risk factor; this datum was found to be significantly correlated with subjects’ having connections in to dental or medical fields (*p* = 0.0032). Moreover, the effects of smoking and alcohol were not identified as important risk factors for developing oral cancer by young Italians.

Poor oral hygiene was identified as a risk factor by 60.59% of the respondents, with significant associations with age (*p* = 0.0021), gender (*p* = 0.0172), and educational attainment (*p* = 0.0121). In detail, males (*p* = 0.0033), junior- (*p* = 0.0000) and high-school (*p* = 0.0124) graduates, and younger participants (*p* = 0.0005) showed higher rates of this answer. Furthermore, a significant association was found between identification of poor oral hygiene as a risk factor and subjects’ connection with patients diagnosed with OSCC (*p* = 0.0312) 

HPV was identified as risk factor by 54.65% of interviewees (including answers c and e).

### 4.5. Features of Oral Cancer Lesions

Generally, the participants showed poor knowledge regarding the early signs and symptoms of oral cancer (Figure 4). In total, only 33.46% identified white plaque and 13.86% identified red plaque as potential early signs of oral cancer. In particular, identification of white lesions was found to be statistically different between subjects responding from Northern Italy compared to those from Southern Italy (*p* = 0.021). Furthermore, only 4.55% of participants identified both white and red plaque as early signs of OSCC. Moreover, 35.44% could not define any related signs of OSSC, and identified only incorrect symptoms.

### 4.6. Habits of Self-Inspection

Only 11.68% of respondents reported knowing how to perform a self-inspection (Figure 5), with a significant association with subjects with connections to dental or medical fields (*p* = 0.003). No statistically significant correlations were found for other groups. 

### 4.7. Time Waited before Seeking Medical Attention

As shown in Figure 6, 84.95% of respondents declared that in the case of an ulcer appearing, they would seek medical attention within 2 weeks, 11.68% within 2 months, and 3.76% within 4 months. Of this latter percentage, 54.45% were younger respondents (*p* = 0.02), showing a significant correlation between waiting time and age (*p* = 0.007). 

### 4.8. Professional Reference Figures

Of the participants, 40.19% would consult a dentist, 48.92% would consult a general practitioner, and 3.36% would consult an otolaryngologist in case of oral mucosal alterations. A further 2.17% would wait and see or use homemade remedies, and 5.34% reported that they would not know who to turn to. No statistical correlations were noted (Figure 7).

### 4.9. Habits of Routine Dental Visits

As shown in Figure 8, 67.12% of the sample reported visiting the dentist at least once a year, with significant associations with regional provenance (*p*= 0.009) and gender (*p*= 0.034). In particular, people from Northern Italy reported making dental visits twice a year (*p* = 0.012) and females reported routine checks once a year (*p* = 0.002). 

## 5. Discussion

This work aimed to survey knowledge about oral cancer among the Italian population. The questionnaire was deliberately anonymous in order to obtain the most unconditioned answers and to collect as many facts as possible. In order to maintain the anonymity of interviewees, no personal data were collected and this decision might have led to a bias: some participants could have answered more than once the questionnaire. However, we considered this issue negligible compared to the possibility of spontaneous answers from interviewees.

The percentage of males and females, different age groups, different provenances (Italian regions), and different educational attainments made the sample representative of the Italian population. Indeed, a coherent distribution of demographic features was observable among the sample. 

In our opinion, the respondents were heterogeneous and well distributed according to age groups and even geographic provenance (as shown in demographic table). This represents a positive factor when drawing a picture of knowledge of population about a topic. The predominance of female respondents in this study (71.49%) was in accordance with other findings [15,16], and it may be explained by random variation or a greater compliance in answering of women. Furthermore, we believe that a 505 person sample was a good starting point for the aim of the study. In future, the sample size could be increased.

Patients’ procrastination in consulting health professionals is the main factor contributing to the delayed diagnosis of OSCC and consequently to its poor prognosis [10]. 

Recent studies reported that “patient delay” is about 1.6–5.4 months [10], and that this could be due to a lack of knowledge about the disease and, thus, to non-recognition of the signs that almost always precede OSCC and that could be easily detected [12]. The results of the present study showed that the level of OSCC knowledge was not satisfactory in our Italian sample. Most of the participants were aged under 50 years; it was found in a work by Panzarella et al (2014) [13] that younger patients show a longer delay with respect to older ones. This datum is probably due to emotional response of younger people, adopting a “wait and see” behavior and denying the usefulness of medical help; thus, they opt for a useless self-diagnosis and/or a self-medication. This should be kept in mind, when evaluating our results. However, in our study, the most significant correlation between waiting time and age was observed for the younger group surveyed, which answered that they would wait a shorter time before seeking medical attention after the appearance of an ulcer. Nevertheless this result might have been due to the size of the Age Group 1 sample compared to the others.

Only 59.60% of participants reported knowing of the existence of OSCC, with remarkable statistical differences observed for all groups. In particular, answers by females were found to be more significant in this respect, as well as those from people with graduate degrees and younger groups. Answers from people with junior-school education were also found to be significantly different; however, these data could be explained by the sample size of the subgroups. Globally, with regard to the existence of oral cancer, these results were almost expected; however, in our opinion, the knowledge could be better-popularized. This issue was supported by results obtained analyzing by answers on the incidence of OSCC. Of the sample, 68.12% reported that every year, in Italy, <3000 new cases of OSCC were diagnosed, representing a considerable underestimation of the disease. Public knowledge of the risk factors is one of the most important factors in successful prevention or early diagnosis in OSCC. Clearly, beyond the epidemiological issue, this datum gives an interesting picture of general knowledge and opinions of oral cancer in Italy, with an important consequence of underestimation and subsequent poor attention to this debilitating disease. Of the 505 answers collected, 4.75% reported having a relative or an acquaintance diagnosed with OSCC; however, this condition seemed not to significantly influence knowledge about the pathology. Indeed, interviewees with relatives diagnosed with OSCC did not report knowing how to do a self-inspection or waiting a shorter time before seeking medical attention, compared to people without a diagnosis of OSCC in the family. They did not differently report features of lesions or risk factors (except for poor oral hygiene), meaning that a family history of oral cancer is not a factor associated with increased knowledge about the disease, even if it affects relatives.

In most OSCC cases, patients have well-known modifiable risk factors, such as tobacco use and alcohol consumption [17,18].This issue deserves to be doubly highlighted, both because a high-risk patient should be checked more often and because of prevention of risk factors. Patients with smoking and/or alcohol habits could report more knowledge of OSCC risk factors, probably because they become more responsive and they join screening programs with an enhanced compliance. Additionally, making risk factors known in younger populations could have a great effect in reducing the spread of habits such as smoking and alcohol use [19,20].

We are confident in the power of knowledge to nudge populations towards better habits, supported by several awareness-raising campaigns that in recent years generated significant results in terms of improving general health [21,22].

In our and other studies [23,24], smoking is widely recognized as a risk factor compared with other potential risk factors, especially alcohol consumption. Further, about half of our respondents indicated alcohol as a risk factor, especially those who declared having acquaintances in medical or dental fields. In the literature, this is well-known data; several studies [25,26,27,28] have reported little public knowledge, reporting that 50% or less of the public know about the risks of alcohol consumption. While we found a good percentage of people recognizing alcohol as a risk in developing OSCC, this information may have merely arisen from other campaigns for general health. This issue could also have been responsible for answers obtained about HPV and its role in the development of OSCC. A low percentage of interviewed reported being informed about the role of HPV in OSCC development.

Cossellu G et al (2019) [29] reported that women with a diagnosis of cervical HPV and their male partners have a high risk for subclinical oral HPV infection. However, as reported in this study, patients disregard the potential links between HPV infection and oral and pharyngeal cancer, and dentists should discuss oral sexual practice with their patients to raise awareness [30].

Poor oral hygiene was identified as a risk factor by 60.59% of the participants, mostly by young participants (Group 1), males, and respondents with junior and high-school diplomas. Of course, this result was unexpected, because poor oral hygiene is not usually reported in information disseminated about oral cancer. The significance of this result could be attributed to a certain intuition of respondents, probably as a consequence of the fact that the questionnaire was sent out by dentists. Therefore, we cannot formulate further hypotheses on corrective actions, only just report results.

In the literature, some surveys from around the world [24,31,32,33,34] have reported low levels of knowledge about the clinical features of OSCC. Interestingly, in our study, one third of participants believed that symptoms as burning or dry mouth were most indicative of oral cancer. Very few participants (13.86%) believed that red lesions might indicate OSCC and only 4.55% identified both white and red plaque; this is particularly impressive, given the high rate of malignant transformation reported in red oral plaque (erythroplakias) [35,36], As a consequence of this misunderstanding of initial signs, patients may not consult a dentist, increasing the duration of diagnostic delay.

Self-inspection was found to be significantly related to early diagnosis of OSCC [37] as well as knowledge about the features of initial oral cancer lesions [34]; therefore, increasing people’s awareness is of course an important goal in OSCC early diagnosis. In this regard, important results have been published concerning other neoplasms, such as breast and testicular cancers and melanoma; the population has more knowledge about self-examination of breasts [38], skin [39], and testicles [40], surely due to massive campaigns of prevention. Our survey respondents did not seem to be informed about self-inspection of the mouth as a preventive measure for OSCC, except for those who had connections with the medical or dental fields. However, having acquaintances is not a sufficient factor to prevent oral cancer; therefore, we foresee the need to emphasize the importance of self-inspection for early diagnosis of OSCC.

Most participants would consult a general practitioner or dental professional if they noticed a lesion in their mouths, but at the same time, only 11.68% of those surveyed reported knowing how to perform a self-inspection and only a few considered red lesions to be a sign of OSCC. Therefore, this answer relies on a false premise. Additionally, 35.44% could not define any related signs of OSSC, only symptoms; in this way, these participants would turn to specialists only in the case that symptoms appeared. From this perspective, correct explanations of initial signs and self-examination of the oral cavity should be provided and strongly encouraged by Italian dentists.

Another consideration that should be made is that only 40.19% of participants reported that they would consult a dental professional in the case of oral lesions, probably because Italian people do not consider oral cancer to be a condition that a dentist can deal with or diagnose. In this case, it could be advisable to better reframe among the population this professional figure as able to play a key role in initial diagnosis and referral.

Similar results were reported for other surveys performed in different countries in Europe [15,16,41,42]; respondents showed poor awareness of and knowledge about oral cancer, especially with regard to modifiable risk factors. In detail, the majority of the European population recognized smoking as a risk factor for oral cancer, whereas alcohol consumption and unprotected intercourse were identified as minor risk factors. Moreover, most European people would tend to visit their general practitioner if they noticed a lesion in their mouths, confirming the need for more awareness about the roles of the dentist and otolaryngologist. Furthermore, these findings clearly underline the importance of extending educational programs in the field of oral cancer to general practitioners [15].

## 6. Future Outlook

In view of all this, we suggest a customization of preventive campaigns, especially taking account of age, risk factors, and the importance of early diagnosis. In our opinion, dental professionals play an essential role in early diagnosis and therefore in reducing the high morbidity and mortality associated with OSCC, both as oral medicine physicians, reducing professional delay, and in cutting down patient delay by increasing their awareness. 

Furthermore, these results might be viewed in epidemiological and environmental frameworks in order to design appropriate intervention planning and increase the knowledge of the population about OSCC. For one thing, dental professionals could introduce into general practice an appropriate anamnestic recording and provide adequate details for their patients, with a recommendation to raise awareness of oral cancer and the feasibility of dentists’ obtaining information. 

Specific programs for improving awareness could be implemented using easy-to-use tools. Indeed, technologies have been found to be effective in increasing the motivation and awareness of patients [43,44], for example as health apps used to discourage incorrect habits and to encourage healthy behaviors and motivate people to work out or to comply with a balanced diet plan. In this regard, something could also be done to nudge the population towards regular self-inspections and dental visits [45]. Of course, technology could serve the purpose of spreading knowledge and making the population more conscious; in this regard, specific and customized actions need to be carried out to teach and nudge Italian young people who do not identify smoking and alcohol as important factors in the development of cancer.

## 7. Conclusions

To conclude, this survey produced a not-encouraging picture of the knowledge of the Italian population about OSCC; therefore, a massive action of sensitization campaigns should be put into place at different levels, beginning from dentists and general physicians and reaching public and state establishments.

## Figures and Tables

**Figure 1 ijerph-17-04889-f001:**
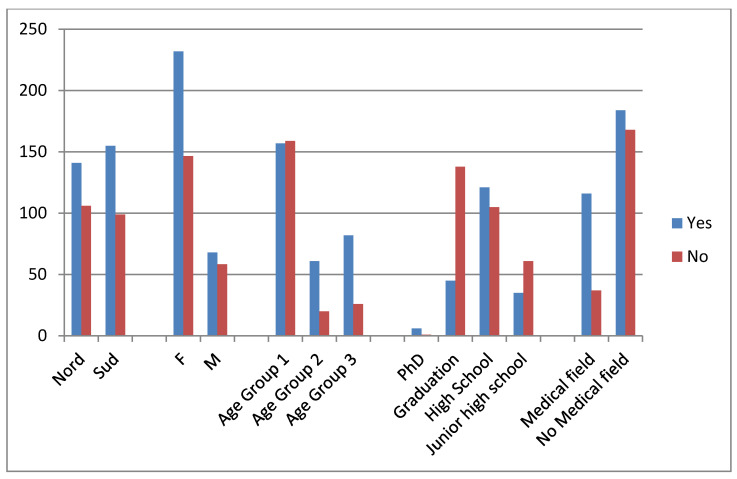
Answers related to knowledge of OSCC.

**Figure 2 ijerph-17-04889-f002:**
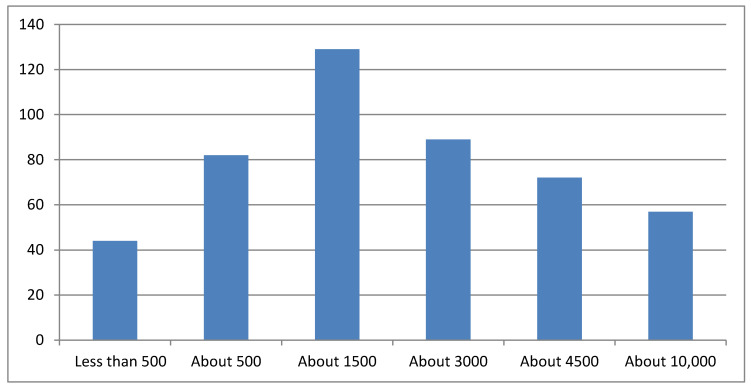
Answers related to cases of OSCC per year in Italy.

**Figure 3 ijerph-17-04889-f003:**
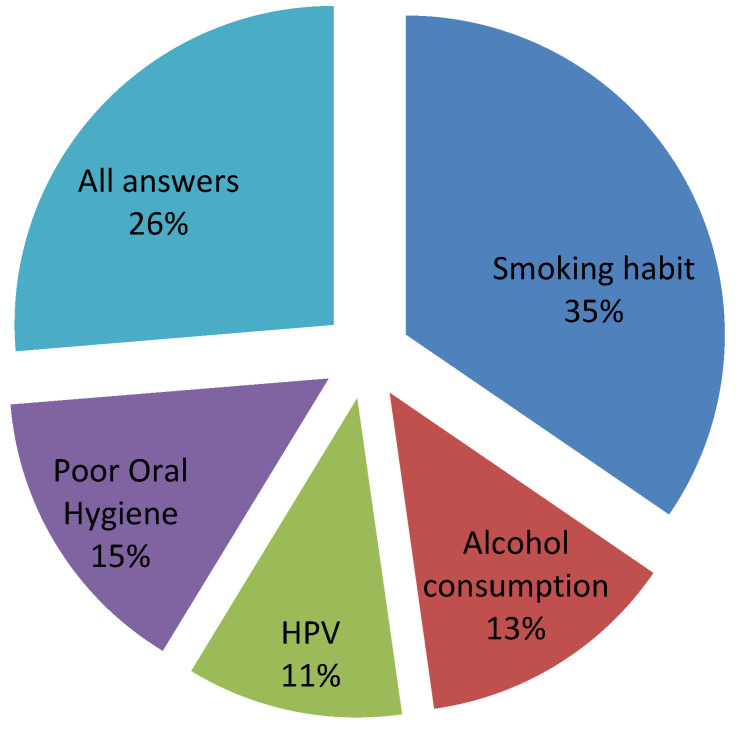
Pie chart of answers related to knowledge of OSCC risk factors.

**Figure 4 ijerph-17-04889-f004:**
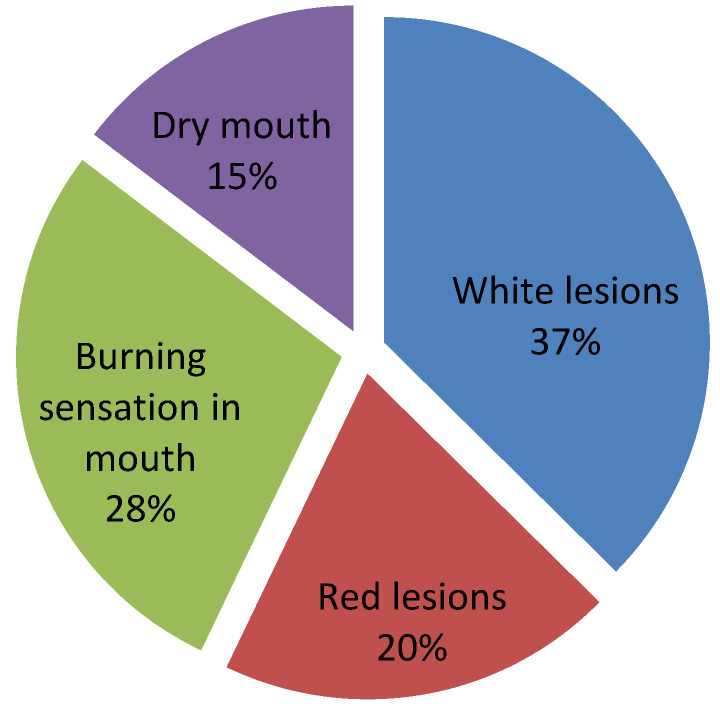
Pie chart of answers related to knowledge of features of OSCC.

**Figure 5 ijerph-17-04889-f005:**
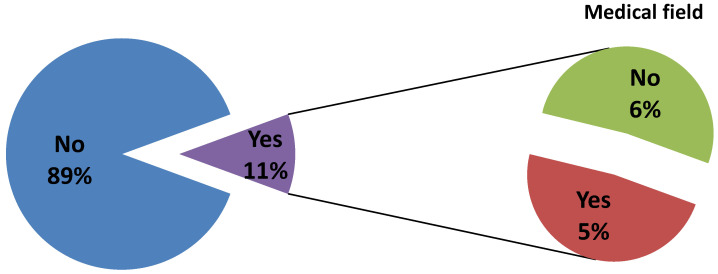
Pie chart of answers related to habits of self-inspection.

**Figure 6 ijerph-17-04889-f006:**
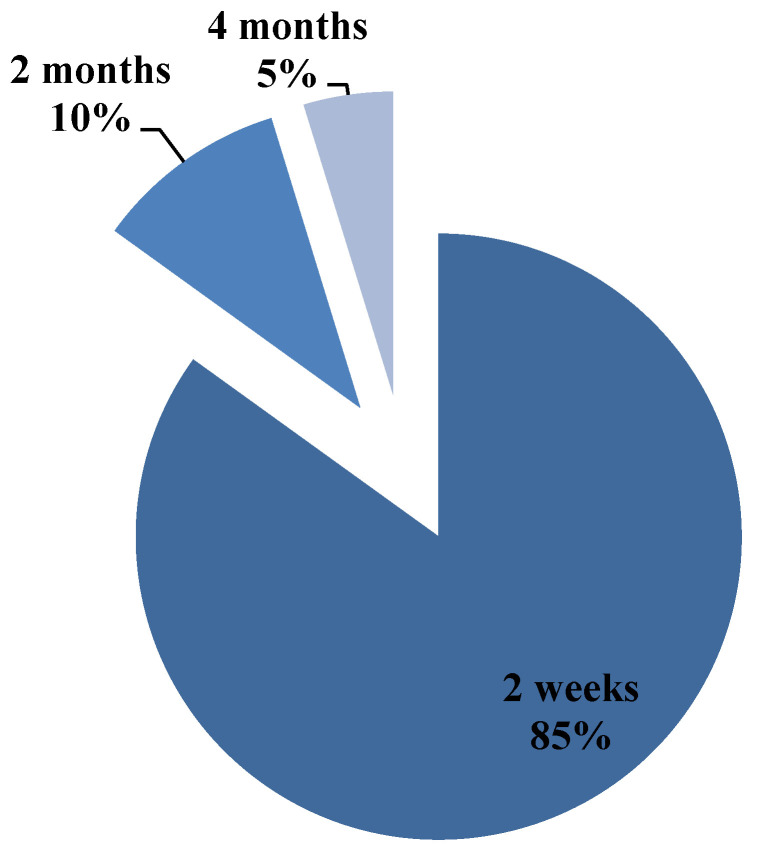
Pie chart of answers related to time waited before seeking medical attention.

**Figure 7 ijerph-17-04889-f007:**
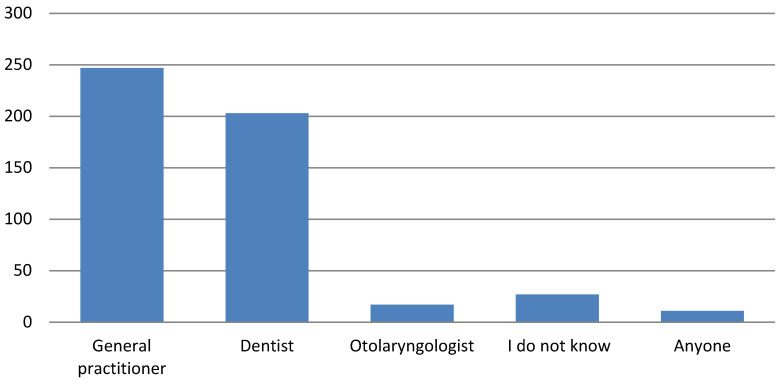
Answers to questions about professional reference figures.

**Figure 8 ijerph-17-04889-f008:**
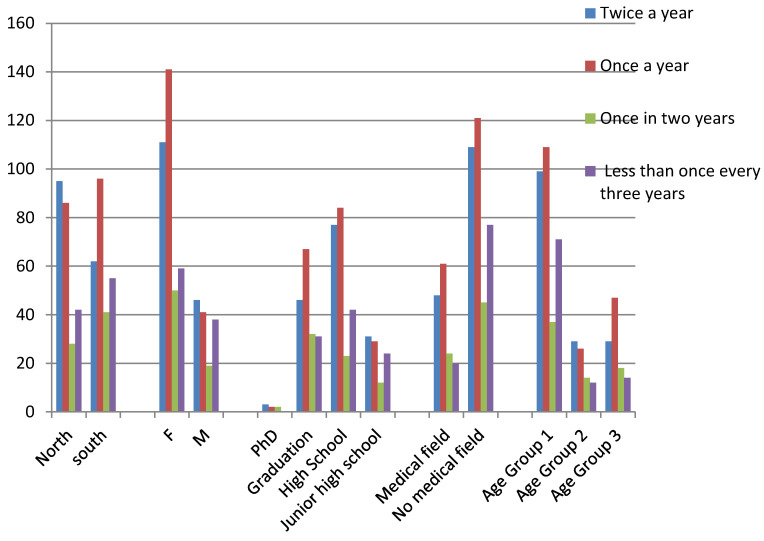
Answers about habits of routine dental visits.

**Table 1 ijerph-17-04889-t001:** Results of statistical analysis.

	Existence of Knowledge	Risk Factors					Signs and Symptoms				Self-Inspection	Referral	Waiting Time	Scheduled Visits
		Smoking	Alcohol	Poor oral Hygiene	HPV	All Risk Factors	White Plaque	Red Plaque	Burning Sensation in Mouth	Dry Mouth				
**Sex**	0	0.844	0.463	0.014	0.102	0.903	0.27	0.707	0.415	0.706	0.119	0.581	0.384	0.034
**Age**	0	0.229	0.341	0.002	0.137	0.759	0.465	0.192	0.487	0.472	0.561	0.109	0.007	0.103
**North/South**	0.456	0.825	0.101	0.41	0.506	0.266	0.019	0.315	0.267	0.459	0.741	0.206	0.163	0.009
**Educational Attainment**	0	0.712	0.816	0.012	0.568	0.428	0.308	0.535	0.43	0.877	0.403	0.192	0.86	0.169
**Medical Relationship**	0	0.914	0.003	0.539	0.7	0.227	0.816	0.11	0.513	0.196	0.002	0.104	0.647	0.119
**Family OSCC**	0	0.945	0.478	0.031	0.22	0.909	0.912	0.608	0.51	0.754	0.119	0.07	0.936	0.081

Statistically significant for *p* ≤ 0.05.

**Table 2 ijerph-17-04889-t002:** Demographic results.

**Gender**	M	144	28.51%
	F	361	71.49%
**Age**	Group 1	316	71.48%
	Group 2	81	16.03%
	Group 3	108	21.38%
**Provenance**	North	251	49.70%
	South	254	50.29%
**Educational Attainment**	Junior high school	101	20%
	High school	217	44.75%
	Graduate degree	180	36.23%
	PhD	7	1.38%
**Relatives with OSCC Diagnosis**	Yes	24	4.75%%
	No	481%	95.24%
**Medical Relationship**	Yes	153	30.29%
	No	352	69.72%

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
