# Peer review of "A Snapshot of Knowledge about Oral Cancer in Italy: A 505 Person Survey"

_ijerph, 2020, doi:10.3390/ijerph17134889_

Round 1
Reviewer 1 Report
The current study described the knowledge of patients around oral cancer (OSCC) within Italy. Performing a web-based survey targeting 505 individuals with a wide range for ages (18 – 76), different questions including existence, incidence, features of lesions, risk factors, self-inspection habits, and preventive actions were highlighted. The assessment was conducted using a Chi-square of independence with adjusted standardized residuals to identify a correlation between different groups. Results revealed that Italian education sectors have to develop a strategic plan to enhance the knowledge of people about the OSCC.
I enjoyed reading this article and only a few minor revisions may further improve the manuscript:
- The article needs reading-proof. Consistency is lacking for fonts, headings, and styles in the current format, and mild English revision is required to communicate.
- Maybe it is a good idea to separate future outlooks from the discussion part to highlight the future direction.
- If possible, compare the situation of Italy to other countries in Europe? Is there any similar relevant study to highlight this?
- As observed, there is a significant bias in the population towards females, how this bias may affect the results of the study?
- It is very interesting to see that the effect of smoking and alcohol were not identified as important sources for developing cancer between different age groups (Table 2). I think the authors must specifically address these challenges for future action targeting young italians.
Author Response
- The article needs reading-proof. Consistency is lacking for fonts, headings, and styles in the current format, and mild English revision is required to communicate.
Done in the text (red highlighted)
- Maybe it is a good idea to separate future outlooks from the discussion part to highlight the future direction.
Done in the text (red highlighted)
- If possible, compare the situation of Italy to other countries in Europe? Is there any similar relevant study to highlight this?
Thank you for your suggestion. Similar results were reported in other surveys performed in different countries in Europe. This aspect was further and better described in the discussion section and red-highlighted in the text
- As observed, there is a significant bias in the population towards females, how this bias may affect the results of the study?
The predominance of female respondents in this study (71,49 %) is in accordance with other findings and it may be explained by random variation. (red-highlighted in the text)
- It is very interesting to see that the effect of smoking and alcohol were not identified as important sources for developing cancer between different age groups (Table 2). I think the authors must specifically address these challenges for future action targeting young Italians.
This issue deserves to be surely carefully explained. Thank you. This aspect was added in results and discussion sections and red-highlighted in the text.
Reviewer 2 Report
Comments:
1. Manuscript is poorly organized and written.
For example: (1) Typing errors in line 37 and 151 (".11. Introduction", ".3 Results").
(2) The "P" in "p- value" should be used in a consistent form (capital or small letter) throughout the manuscript.
2. Graph 1 is confusing and hard to interpret. Moreover, its a pie chart not a graph.
3. Authors are advised to add more figures/graphs to make it easy to follow and to strengthen the manuscript.
Author Response
- Manuscript is poorly organized and written. For example: (1) Typing errors in line 37 and 151 (".11. Introduction", ".3 Results"). (2) The "P" in "p- value" should be used in a consistent form (capital or small letter) throughout the manuscript.
a complete revision was performed (red highlighted)
- Graph 1 is confusing and hard to interpret. Moreover, its a pie chart not a graph.
Thank you for this indication. We modified Graph 1
- Authors are advised to add more figures/graphs to make it easy to follow and to strengthen the manuscript.
Thank you for this indication. We added Graphs for all answers evaluated.
Reviewer 3 Report
Dear editor,
you addressed a very interesting topic. However, I have some doubts about the study design:
- did you submit the study project to the Ethic Committee?
- was your questionnaire validated?
- how did you recruit the subjects?
- how did you choose the subjects?
- were the subjects representative of the Italian population?
Author Response
1) did you submit the study project to the Ethic Committee?
Present study is not invasive intervention and it is anonymous, because of this, we did not believe necessary to submit them to Ethic Committee. Further, the survey design is an easy-to-use way to investigate and it is often used to assess general features of population.
2) was your questionnaire validated?
A pilot test of questionnaire was performed among a random sample of 10 patients referred to our department (on a volunteer basis) to ensure practicability, validity, and interpretation of answers. Then, the questions were revised considering the comments obtained before sharing the link of web-based questionnaire to the study sample. (written in the material and methods section and red-highlighted)
Furthermore, authors responsible for designing the study (NR, CG) checked critical points and managed them in order to make clearer the questions. The final approval was gave by all authors before to submit it to sample. (written in the material and methods section and red-highlighted)
3) how did you recruit the subjects?
Questionnaire was sent to dental practitioners asking them for sharing it with patients and to general population by URL sharing, by building on knowledge of all authors and colleagues. (Red in the text)
4) how did you choose the subjects?
No exclusion criteria were stated in submitting and sending questionnaire. In collecting answers and analysing data, only answers from subjects belonging to medical field were excluded in order to have a picture of laypeople and therefore avoid selection bias. (This aspect was better clarified in material and methods section and red-highlighted in the text)
5) were the subjects representative of the Italian population?
In our opinion, the respondents were heterogeneous and well distributed according to age-groups and even geographic provenance (as in demographic table). This represents a plus point useful to draw a picture of knowledge of population about a topic. Furthermore, we believe that a 505 people sample could be a good starting point for the aim of the study. Surely, the sample size could be more increased. (This aspect was better clarified in discussion section and red-highlighted in the text)
Round 2
Reviewer 2 Report
Minor comment:
Graphs are too basic, authors are advised to improve them.
Author Response
- Graphs are too basic, authors are advised to improve them.
Graphs were improved
Reviewer 3 Report
Dear authors, thank you for your replies. This is an observational study and approval by Ethic Committee is required. The population study is not representative of the Italian population (you do not have any data ti support your statements about this point) and the questionnaire has not been validated.
Author Response
- Dear authors, thank you for your replies. This is an observational study and approval by Ethic Committee is required. The population study is not representative of the Italian population (you do not have any data to support your statements about this point) and the questionnaire has not been validated.
The study protocol and the questionnaire as well were approved by the review board of Dental School of Verona (University of Verona, Italy). review board of Dental School of Verona (University of Verona, Italy). Ethical Committee approval was not asked because the respondents and their personal data were protected by anonymity. Not even the person who analyzed data could link information to somebody, furthermore, people could not be recognized by answers given. (Written in the material and methods section and red-highlighted).
The questionnaire was shared to people by e-mail, WhatsApp™ and Telegram Messenger chat room and Facebook. This choice aimed to reach as more people as possible.
The percentage of male and female, different age-groups and different provenance (Italian regions), different educational attainments make the sample representative of Italy. Indeed, a coherent distribution of people features among the sample was observable. The percentage of female is slightly higher than that of male, probably due to more compliance in answering of women. (written in the discussion section and red-highlighted)
The validity of a questionnaire is determined by analyzing whether the questionnaire measures what it is intended to measure. The content validity was assessed by a panel of Researchers of Dental School of Verona (review Board). (written in the material and methods section and red-highlighted).
The content validity was not considered needed because it was impossible to ask questions in another way or in preexisting ways. Anyway, from using the first questionnaire (the pilot questionnaire) it was possible to assess whether the respondents judged the questionnaire items to be valid.
For better clarifying, we report the paper on which we based to design our questionnaire. Tsang S, Royse CF, Terkawi AS. Guidelines for developing, translating, and validating a questionnaire in perioperative and pain medicine. Saudi J Anaesth. 2017;11(Suppl 1):S80-S89. doi:10.4103/sja.SJA_203_17